# Human Milk from Previously COVID-19-Infected Mothers: The Effect of Pasteurization on Specific Antibodies and Neutralization Capacity

**DOI:** 10.3390/nu13051645

**Published:** 2021-05-13

**Authors:** Britt J. van Keulen, Michelle Romijn, Albert Bondt, Kelly A. Dingess, Eva Kontopodi, Karlijn van der Straten, Maurits A. den Boer, Judith A. Burger, Meliawati Poniman, Berend J. Bosch, Philip J. M. Brouwer, Christianne J. M. de Groot, Max Hoek, Wentao Li, Dasja Pajkrt, Rogier W. Sanders, Anne Schoonderwoerd, Sem Tamara, Rian A. H. Timmermans, Gestur Vidarsson, Koert J. Stittelaar, Theo T. Rispens, Kasper A. Hettinga, Marit J. van Gils, Albert J. R. Heck, Johannes B. van Goudoever

**Affiliations:** 1Department of Pediatrics, Amsterdam UMC, Vrije Universiteit, University of Amsterdam Emma Children’s Hospital, 1105 AZ Amsterdam, The Netherlands; b.j.vankeulen@amsterdamumc.nl (B.J.v.K.); m.romijn1@amsterdamumc.nl (M.R.); e.kontopodi@amsterdamumc.nl (E.K.); d.pajkrt@amsterdamumc.nl (D.P.); a.schoonderwoerd@amsterdamumc.nl (A.S.); 2Biomolecular Mass Spectrometry and Proteomics, Bijvoet Center for Biomolecular Research and Utrecht Institute for Pharmaceutical Sciences, University of Utrecht, 3584 CH Utrecht, The Netherlands; a.bondt@uu.nl (A.B.); k.a.dingess@uu.nl (K.A.D.); m.a.denboer@uu.nl (M.A.d.B.); m.hoek@uu.nl (M.H.); s.tamara@uu.nl (S.T.); a.j.r.heck@uu.nl (A.J.R.H.); 3Netherlands Proteomics Center, Padualaan 8, 3584 CH Utrecht, The Netherlands; 4Food Quality & Design Group, Wageningen University and Research, 6708 WG Wageningen, The Netherlands; kasper.hettinga@wur.nl; 5Department of Medical Microbiology, Amsterdam UMC, University of Amsterdam, 1105 AZ Amsterdam, The Netherlands; k.vanderstraten@amsterdamumc.nl (K.v.d.S.); j.a.burger@amsterdamumc.nl (J.A.B.); m.poniman@amsterdamumc.nl (M.P.); p.j.brouwer@amsterdamumc.nl (P.J.M.B.); r.w.sanders@amsterdamumc.nl (R.W.S.); m.j.vangils@amsterdamumc.nl (M.J.v.G.); 6Division Infectious Diseases & Immunology/Laboratory of Virology, Department Biomolecular Health Sciences, Faculty of Veterinary Medicine, Utrecht University, Yalelaan 1, 3584 CL Utrecht, The Netherlands; b.j.bosch@uu.nl (B.J.B.); W.Li@uu.nl (W.L.); 7Department of Obstetrics and Gynaecology, Amsterdam UMC, Vrije Universiteit, 1081 HV Amsterdam, The Netherlands; cj.degroot@amsterdamumc.nl; 8Department of Microbiology and Immunolgy, Weill Medical College of Cornell University, 1300 York Avenue, New York, NY 10065, USA; 9Wageningen Food & Biobased Research, Wageningen University and Research, P.O. Box 17, 6700 AA Wageningen, The Netherlands; rian.timmermans@wur.nl; 10Department of Experimental Immunohematology, Sanquin Research, Landsteiner Laboratory, Amsterdam UMC, University of Amsterdam, P.O. Box 9190, 1006 AD Amsterdam, The Netherlands; G.Vidarsson@sanquin.nl; 11Viroclinics Xplore, Viroclinics Biosciences B.V., Nistelrooise Baan 3, 5374 RE Schaijk, The Netherlands; stittelaar@viroclinics.com; 12Department of Immunopathology, Sanquin Research & Landsteiner Laboratory Academic Medical Centre, 1081 HV Amsterdam, The Netherlands; T.Rispens@sanquin.nl

**Keywords:** immunoglobulins, pasteurization, COVID-19, breastfeeding

## Abstract

Background: Since the outbreak of coronavirus disease 2019 (COVID-19), many put their hopes in the rapid availability of effective immunizations. Human milk, containing antibodies against syndrome coronavirus 2 (SARS-CoV-2), may serve as means of protection through passive immunization. We aimed to determine the presence and pseudovirus neutralization capacity of SARS-CoV-2 specific IgA in human milk of mothers who recovered from COVID-19, and the effect of pasteurization on these antibodies. Methods: This prospective case control study included lactating mothers, recovered from (suspected) COVID-19 and healthy controls. Human milk and serum samples were collected. To assess the presence of SARS-CoV-2 antibodies we used multiple complementary assays, namely ELISA with the SARS-CoV-2 spike protein (specific for IgA and IgG), receptor binding domain (RBD) and nucleocapsid (N) protein for IgG in serum, and bridging ELISA with the SARS-CoV-2 RBD and N protein for specific Ig (IgG, IgM and IgA in human milk and serum). To assess the effect of pasteurization, human milk was exposed to Holder (HoP) and High Pressure Pasteurization (HPP). Results: Human milk contained abundant SARS-CoV-2 antibodies in 83% of the proven cases and in 67% of the suspected cases. Unpasteurized milk with and without these antibodies was found to be capable of neutralizing a pseudovirus of SARS-CoV-2 in (97% and 85% of the samples respectively). After pasteurization, total IgA antibody levels were affected by HoP, while SARS-CoV-2 specific antibody levels were affected by HPP. Pseudovirus neutralizing capacity of the human milk samples was only retained with the HPP approach. No correlation was observed between milk antibody levels and neutralization capacity. Conclusions: Human milk from recovered COVID-19-infected mothers contains SARS-CoV-2 specific antibodies which maintained neutralization capacity after HPP. All together this may represent a safe and effective immunization strategy after HPP.

## 1. Introduction

The severe acute respiratory syndrome coronavirus 2 (SARS-CoV-2) outbreak, which was first reported in December 2019, has had an enormous global impact. SARS-CoV-2 can cause coronavirus disease 2019 (COVID-19) with the number of confirmed cases over 130 million, and over 2.8 million deaths globally as of April 2021.

In response to the pandemic, many countries have had to introduce drastic lockdowns to enforce physical separation, affecting economies worldwide, while also imposing a huge psychological burden on specific groups such as the elderly and school-aged children. On a personal level, general preventive measures like protective materials, physical distancing and frequent hand washing, have shown to be effective. As these measures are not sustainable for prolonged periods of time, the pandemic has necessitated rapid development of effective vaccines as prevention. Even with the development of several COVID-19 vaccines and extensive vaccination efforts, we are still far from global vaccination goals [1].

When looking at preventive strategies, it is interesting to note that, in infants, breastfeeding is associated with a 30% reduction in respiratory infections when compared to formula feeding [2,3]. It is generally accepted that this protective effect is due to the human immune components in human milk, such as specific antibodies of which secretory immunoglobulin A (sIgA) is the most abundant. SIgA represents our first line of defense as it acts directly at mucosal surfaces [4]. SIgA inhibits microbial binding to host receptors of intestinal epithelial cells, entrapping pathogenic microorganisms within the mucus and enhancing ciliary activities, thus eliminating invading pathogens [5,6,7]. Through this mechanism, human milk sIgA may provide protection against entry of SARS-CoV-2 in the airway at mucosal surfaces.

The structural proteins of SARS-CoV-2 include the spike (S), nucleocapsid (N), membrane, and envelope proteins (Figure 1). The S1 subunit of the S protein contains the receptor binding domain (RBD), which facilitates angiotensin-converting enzyme 2 (ACE2) receptor mediated virus attachment, while the S2 subunit of the S protein promotes membrane fusion to initiate the infection of host cells. The N protein encapsulates viral RNA and is necessary for viral transcription and replication [8]. The human immune system will, when infected by SARS-CoV-2, generate antibodies against one or more of these viral proteins, whereby variability may exist in the immunoglobulin class preferentially made (e.g., IgG, IgA, IgM) and the antigen to which the immunoglobulin binds. The titers of antibodies are individual specific, but also amongst others determined by the severity of the infection and the time that has passed since the onset of the infection. Therefore, it is recommended to use complementary assays to determine SARS-CoV-2 antibody titers.

There is strong evidence that antibodies, especially of the IgA class, against several respiratory infections, such as influenza, are secreted into human milk [9,10]. A previous study indicated that 15–30 days following the onset of symptoms, antibodies against the SARS-CoV-2 RBD may be present in human milk [11]. These data form the basis of our hypothesis that an array of SARS-CoV-2-reactive antibodies may be present in human milk from mothers who have recovered from COVID-19. Human milk sIgAs may provide insights into clinical strategies to reduce the incidence of SARS-CoV-2 infections, by neutralizing the virus in the airway mucosa, although many steps need to be taken before such an approach can be implemented. This is an intriguing perspective as monoclonal antibody therapies can provide a means of treatment for the disease even after a vaccine is available, particularly in vulnerable populations. However, human milk may contain pathogens, and therefore pasteurization is required prior to use [12]. Holder pasteurization (HoP), a heat treatment at 62.5 °C for 30 min, is currently the standard pasteurization method for human milk [13]. Although HoP effectively inactivates microbial contaminants, it concomitantly reduces the activity of some important bioactive milk components [13,14]. To prevent such reduced activity, alternative methods to HoP, such as high pressure pasteurization (HPP), are currently being investigated [15,16]. Our aim was thus to evaluate the level of SARS-CoV-2 reactive antibodies and determine efficacy of virus neutralization in serum, unpasteurized human milk, and in human milk after thermal (HoP) and non-thermal (HPP) pasteurization.

## 2. Materials and Methods

### 2.1. Study Population

This prospective case control study aimed to include 40 lactating women with a confirmed or high probability of COVID-19. Lactating women who recovered from a proven COVID-19 infection were recruited by an online recruitment letter. A confirmed infection was defined as a positive SARS-CoV-2 PCR from a nasal-pharyngeal swab. Subjects were classified in the suspected COVID-19 group in the event of a confirmed infection with SARS-CoV-2 in the household and if the lactating woman developed COVID-19 symptoms. A control group, of 15 healthy lactating women, was recruited simultaneously as the proven COVID-19 infected group from the Amsterdam UMC if they met the following criteria: lactating women who delivered at Amsterdam UMC with a negative SARS-CoV-2 PCR from a nasal-pharyngeal swab during delivery, and without symptoms of COVID-19. Ethical approval was obtained from the Medical Ethics Committee of the Amsterdam UMC, location VUmc and written informed consent was obtained from all participants.

### 2.2. Material Collection

All participants were requested to collect 100 mL of human milk in specially provided bottles and to store the bottle in their freezer until collected by study staff during a home visit. Subsequently, the samples were stored at −20 °C. During the home visit, maternal serum was collected by a trained phlebotomist.

### 2.3. Laboratory Analyses

#### 2.3.1. Evaluation of Antibodies in the Serum and Human Milk

To assess the diversity and variability of antibodies present in human milk and serum we decided to use multiple complementary assays. First, ELISA with the SARS-CoV-2 spike protein to detect specific IgA and IgG in human milk and serum, respectively. Second, ELISA with the SARS-CoV-2 receptor binding domain (RBD) and with the SARS-CoV-2 nucleocapsid (N) protein for specific IgG (serum only). Third, a bridging ELISA with the SARS-CoV-2 RBD and N protein for specific total Ig (IgG, IgM and IgA in human milk and serum). All of the ELISA based assays used are depicted in Figure 1 and described in more detail below.

#### 2.3.2. Detection of Anti-SARS-CoV-2 Ig in Serum and Human Milk with ELISA

Soluble prefusion-stabilized S-protein of SARS-CoV-2 were generated as previously described [17]. This protein was immobilized on a 96-well plate (Greiner, Kremsmünster, Austria) at 5 µg/mL in 0.1 M NaHCO_3_ overnight, followed by a one-hour blocking step with 1% casein PBS (Thermo Scientific, Waltham, MA, USA). Human milk was diluted 1:5 and serum was diluted 1:100 in 1% casein PBS and incubated on the S-protein coated plates for 2 h to allow binding. Antibody binding was measured using 1:3000 diluted HRP-labeled goat anti-human IgG (Jackson Immunoresearch, West Grove, PA, USA) in casein for the serum samples and 1:3000 diluted HRP-labeled goat anti-human IgA (Biolegend, San Diego, CA, USA) in casein for the human milk samples. The healthy controls (serum and human milk) were used to determine cut-off values defined as the mean plus two times the standard deviation. Specificity of the ELISA was shown to be >95% for both serum and human milk and the sensitivity was >90% for serum and >80% for human milk.

#### 2.3.3. Bridging ELISA with the SARS-CoV-2 RBD and Nucleocapsid Protein

Antibodies against RBD protein were measured as total Ig (IgG, IgA and IgM) ELISA and an IgG ELISA as described previously [18]. Briefly, for total antibodies, samples were incubated (1:10 for serum, undiluted for human milk) on plates coated with RBD protein (produced in-house [18]) and specific antibodies were subsequently detected using biotinylated RBD protein (produced in-house [18]). For the IgG ELISA, serum samples were diluted 1:100 and incubated on RBD-coated plated, followed by detection of specific IgG antibodies using a mouse monoclonal anti-human IgG antibody (produced in-house [18]). Total Ig against N protein in serum was measured in 1:10 diluted serum on N protein-coated plates followed by detection using biotinylated N protein. In order to determine the cut-off values, pre-pandemic controls were used to provide ∼99% specificity as previously described [18]. The results of the pre-pandemic controls are not described in this paper.

#### 2.3.4. Effect of Antibodies on Virus Replication

##### Pseudovirus Neutralization Assay

Neutralization assays and the generation of a SARS-CoV-2 pseudovirus containing a NanoLuc luciferase reporter gene were performed as previously described [19]. Briefly, HEK 293T cells (ATCC, CRL-11268) were transfected with a pHIV-1NL43∆ENVNanoLuc reporter virus plasmid and a SARS-CoV-2-S∆19 plasmid. Cell supernatant containing the pseudovirus was harvested 48 h post transfection, centrifuged for 5 min at 500× *g* and sterile filtered through a 0.22 µm pore size PVDF syringe filter. For neutralization assays HEK 293T expressing the SARS-CoV-2 receptor ACE2 (HEK 293T/ACE2 [19]) were cultured in DMEM (Gibco), supplemented with 10% fetal bovine serum (FBS), penicillin (100 U/mL), and streptomycin (100 µg/mL).To determine the neutralization activity in serum or human milk, HEK 293T/ACE2 cells were first seeded in 96-well plates coated with 50 µg/mL poly-l-lysine at a density of 2 × 104/well in the culture medium as described above, but with GlutaMax (Gibco) added. The next day, duplicate serial dilutions of heat inactivated serum or human milk samples were prepared in the same medium as used for seeding of cells and mixed 1:1 with ∼1 × 10^3^ infectious units pseudovirus. This mixture was incubated at 37 °C for 1 h before adding it to the HEK 293T/ACE2 cells in a 1:1 ratio with the cell culture medium. After 48 h, the cells were lysed and luciferase activity was measured in the lysates using the Nano-Glo Luciferase Assay System (Promega, Madison, WI, USA). Relative luminescence units (RLU) were normalized to those from cells infected with SARS-CoV-2 pseudovirus in the absence of sera/saliva/swabs. Neutralization titers (ID50-values) were determined as the serum dilution at which infectivity was inhibited by 50%.

##### Replication Inhibition of a SARS-CoV-2 Clinical Isolate Assay

In order to assess if human milk with SARS-CoV-2 specific antibodies possess virus neutralizing activity, in vitro neutralizing assays were conducted using a SARS-CoV-2 clinical isolate strain, which was kindly provided by Christian Drosten, Charité-Universitätsmedizin, Berlin, Germany (BetaCoV/Munich/BavPat1/2020), performed under biosafety level 3+ conditions. In brief, 60 μL of SARS-CoV-2 working dilution containing approximately 200 TCID50/well was mixed with 60 μL of serially 2-fold dilutions of heat-inactivated serum or milk, in triplicates and incubated for 60 min at 37 °C to allow for neutralization of the virus. Subsequently 100 μL of these virus/antibody mixtures were added to confluent VERO E6 cell monolayers (ATCC; CRL-1586) and incubated at 37 °C for four to six days. The virus working dilution and the original virus stock were titrated in a parallel plate and served as positive virus controls in each assay run. After incubation at 37 °C 20 μL of a WST-8 Cell Counting Kit-8 (CCK-8) solution (Sigma-Aldrich, St. Louis, MO, USA; 96992) was added to each well of the plate, followed by an incubation for three hours at room temperature. The absorbance at 450 nm was measured using microplate reader (Synergy H1, Biotek, Winooski, VT, USA). The Reed and Muench method was used to determine the 50% end-point titer of the sample, as well as the virus titer (stock and back titration).

#### 2.3.5. Evaluation of the Effect of Pasteurization of Human Milk on SARS-CoV-2 Antibodies

To assess the effect of pasteurization we used two methods of pasteurization on all of the collected human milk samples, after treatment we evaluated the amount of SARS-CoV-2 antibodies and neutralizing capacity of human milk between raw milk and pasteurized milk samples. During Holder pasteurization (HoP), using current standard methods, human milk is pasteurized at 62.5 °C for 30 min. An alternative to HoP pasteurization is high pressure pasteurization (HPP), which inactivates vegetative (including pathogenic) micro-organisms, yeasts, molds and viruses, without causing heat-induced damage [20].

Samples were stored frozen at −20 °C and thawed overnight in a refrigerator (7 °C), prior to being transferred into sterile pouches that were double packed and treated in a pilot-scale high-pressure unit with water at ambient temperature [21]. We applied a hydrostatic pressure of 500 MPa for 5 min. All samples were stored at −20 °C directly after treatment.

#### 2.3.6. Monitoring IgA Clone Diversity in Human Milk by Mass Spectrometry

In addition to the classical antibody detection assays, we used a novel mass spectrometry (MS) method to examine the IgA clonal diversity in human milk. We examined the IgA clones in unpasteurized human milk, and after the two different pasteurization techniques. The antigen binding fragments (Fab) were proteolytically released from the captured IgAs, and the resulting Fab fragments (45–50 kDa) of individual clones were profiled using MS. The abundance of each unique detected clone could be determined, and thus for each clone the effect of the two different pasteurization techniques could be monitored.

Stabilized spike protein from SARS-CoV-2 was produced and purified as described before [22]. The Spike protein was coupled to NHS-activated agarose for two hours at room temperature. Free NHS groups were inactivated by incubation with 1 M Tris for 30 min. Spike recognizing molecules were captured from milk by incubating 250 µL unpasteurized milk and 100 µL PBS for 2 h end-over-end. The non-binding fraction was collected by centrifugation, and the SARS-CoV-2 specific antibodies were eluted using 100 mM Glycine-HCl (pH 2.7) and immediately neutralized with 1 M Tris (pH 8.0). Next, the IgA antibodies were captured on CaptureSelect IgA affinity matrix from the neutralized eluate (substituted with blocking powder, final concentration 1%), from 140 µL of the non-binding fraction, or from 100 µL unpasteurized milk. Then, Fab portions were generated as described above, collected by centrifugation, and immediately measured on the mass spectrometer.

### 2.4. Statistical Analysis

Patient characteristics and COVID-19 symptoms were expressed as mean with standard deviation (SD) or median with interquartile range (IQR) depending on their distribution. Statistical analysis was performed with IBM SPSS Statistics for Windows, version 26 (IBM Corp., Amonk, NY, USA). In order to compare SARS-CoV-2 IgA in unpasteurized milk of the cases and controls, a Mann-Whitney U test was performed in Graphpad Prism 8.2.1. In order to evaluate the effect of pasteurization on antibody level and neutralization capacity, statistical analyses were performed, depending on the distribution of the data. A Wilcoxon matched-pairs signed rank test was performed in order to compare IgA retention according to LC/MS profiles following HoP and HPP and to compare Spike IgA titers between HPP and HoP milk as a % relative to UP. These tests were performed in Python 3.8.8, Pandas 1.2.3, Numpy 1.19.2, Scipy 1.6.1, visualized with Matplotlib 3.3.4 and Seaborn 0.11.1. Differences in neutralization capacity of the pseudovirus of human milk after pasteurization was tested with a Friedman test in SPSS Statistics for Windows, version 26. To test which groups differ, a Dunn’s post hoc test was performed. Differences in neutralization capacity of the pseudovirus between the confirmed cases, suspected cases and controls was tested with a Kruskal Wallis test in SPSS Statistics for Windows.

## 3. Results

Our prospective case control study included 40 lactating women with confirmed or a high probability of COVID-19 and 15 healthy controls during pandemic. Four women had active COVID-19 symptoms on the day of the scheduled house visit, therefore we were not able to collect the body materials of these women, resulting in a final study population of 38 cases and 13 controls. Three of the subjects with a confirmed infection were admitted to the hospital. For all subjects, samples were obtained at different time intervals from the onset of clinical symptoms, as shown in Table 1.

### 3.1. SARS-CoV-2 Antibodies in Human Milk and Serum

Human milk contained antibodies against the SARS-CoV-2 virus, using any of the assays, in 24 out of 29 (83%) proven cases and in six out of nine (67%) of the suspected cases (Figure 2). A large variability in antibody levels was found in the milk samples of all subjects. Both the assay assessing IgA response against the S protein and the assay detecting the total Ig response against RBD showed a variable pattern in antibody type and the SARS-CoV-2 protein it recognized (Appendix A). The median and range of SARS-CoV-2 Spike-IgA and RBD-Ab in unpasteurized milk from the case and control groups (*p* < 0.001) are depicted in Figure 3.

With a complementary method, using the spike protein from SARS-CoV-2 to enrich for antibodies from milk, analyzing their Fab fragments subsequently by LC-MS, we were able to demonstrate the presence of a few high abundant SARS-CoV-2 antigen-specific antibodies in milk of COVID-19 recovered women (Figure 4).

Antibody secretion may well be dependent on the time that has passed since the onset of COVID-19 and the severity. By using a cross sectional sampling design, we were able to show that even up to 13 weeks from disease onset, detectable levels of antibodies were found in both human milk and serum (Figure 5). While over 80% of the human milk samples in the PCR proven cases contained antibodies, all of their blood samples showed a positive response in at least one of the assays.

### 3.2. Pasteurization of Human Milk and IgA Antibodies

Following the detection of antibodies in milk, we aimed to quantify the effect of different pasteurization methods on the antibody levels. Using mass spectrometry techniques we noticed that following HoP and HPP, the relative abundance of each clone remained largely unaffected, although the actual concentration of most clones, and thus of total IgA, was slightly lowered for both methods of pasteurization relative to UP milk samples (Figure 6a). The reduction in overall IgA concentration was greater in HoP- than HPP-treated milk. In contrast, we did observe a significant difference in the levels of IgA anti S protein antibodies after both methods of pasteurization, with lower levels in HPP milk compared to HoP milk (Figure 6b and Appendix A).

### 3.3. SARS-CoV-2 Virus Neutralization in Unpasteurized and Pasteurized Milk

Next, we aimed to assess if human milk that contained antibodies against SARS-CoV-2 was able to reduce virus replication using two different models. First, neutralization of a SARS-CoV-2 pseudovirus was determined for both serum and human milk (Figure 2 and Figure 7). In both cases (proven and suspected) and controls, neutralization capacity was observed in unpasteurized milk, although, the median of the ln transformed neutralizing capacity, defined as 50% inhibitory dilution, was comparable in milk from women with proven COVID-19 2.36 (IQR 2.03–2.53) and milk from women with a suspected infection 2.37 (IQR 1.96–2.39) and higher compared with controls 1.75 (IQR 1.39–2.19) (*p* = 0.009). Neutralization capacity differed between unpasteurized and pasteurized milk (*p* < 0.0001) and was generally better preserved after the HPP (*p* = 0.906). In contrast, after HoP a substantial decrease in neutralization capacity was observed (*p* < 0.001).

Second, using a replication inhibition with a SARS-CoV-2 clinical isolate, the inhibitory capacity of human milk was determined (Appendix A). In seven out of the 38 (18%) unpasteurized milk samples, and in eight out of the 38 (21%) HPP milk samples, neutralization capacity was observed. None of the HoP milk samples showed neutralization capacity. Some (*n* = 3) HPP milk samples did show neutralizing capacity while the same sample before pasteurization did not exert such an effect. We could not detect a linear correlation between milk antibody levels and virus neutralizing capacity.

All these data together indicate that heat treatment may have caused a loss-of-function even though this did not result in reduced antibody levels according to the preformed ELISA assays. This shows the importance of testing the functionality of the milk in a neutralization assay in addition to the analysis of the antibody levels.

## 4. Discussion

We demonstrate that human milk of mothers who recovered from COVID-19 contains significant amounts of IgA against SARS-CoV-2, for at least 13 weeks following the onset of COVID-19 symptoms. After pasteurization, total IgA antibody levels were affected by HoP, while SARS-CoV-2 specific antibody levels were affected by HPP. Pseudovirus neutralizing capacity of the human milk samples was only retained with the HPP approach. No correlation was observed between human milk antibody levels and neutralization capacity.

We also show that human milk samples from some donors had neutralizing capacity against a SARS-CoV-2 clinical isolate in the stringent replication inhibition assay, resulting in significant inhibition of virus propagation, while most donors had human milk and serum neutralizing capacity against a SARS-CoV-2 pseudovirus. Interestingly, neutralizing capacity was also observed in human milk samples of the control group, which can be explained by the presence of other antiviral proteins, besides antibodies [23]. Only few human milk samples with SARS-CoV-2 antibodies were able to neutralize the SARS-CoV-2 clinical isolate, which could be due to high dilution of the samples, which was necessary in order to prevent cell toxicity induced by human milk. However, the sometimes conflicting results (neutralization capacity of three HPP treated milk samples and not of the same unpasteurized milk samples) in the neutralizing test using the clinical isolate demonstrates a lack of robustness for this test in our setting. It seems unlikely that the HPP procedure adds neutralizing capacity. Another factor that has to be taken into account as an explanation for the different results from these neutralization assays, is that the clinical isolate assay tests replication inhibition of SARS-COV-2, in contrast to the pseudovirus assay tests the prevention of SARS-CoV-2 infection.

Our data clearly indicate strong variability in an individual’s antibody levels in response to COVID-19 infection. Some subjects show a more N protein or RBD directed antibody response, and some exhibit a stronger IgA or IgG response. The presence and abundance of SARS-CoV-2 specific antibodies is known to be variable, for instance, the IgG response to the N protein is believed to occur earlier than the response to the S protein, but the titers are generally lower to the N protein compared to the S protein [24]. Also, in general IgM and IgA-class responses often occur earlier following disease onset, while IgG responses occur later and seem to be longer lasting [25]. Together, these results imply that testing for the presence of antibodies should not be directed against a single viral protein or focus on a single antibody class, but that different proteins and classes should be targeted to obtain the most complete and reliable information.

Donor milk banks around the world use HoP as a way to provide donor milk to preterm and sick term infants. However, HoP is known to affect the immune protection provided by human milk, due to the heat load which the milk is exposed. One promising alternative to HoP is HPP. HPP is already widely used in food industries as a non-thermal food preservation method that provides microbiologically safe products, while at the same time reducing the heat-induced damage of regular thermal pasteurization. Recent studies on HPP of donor human milk indicate that this method is capable of retaining significantly higher levels of antibodies when compared to HoP, while at the same time successfully eliminating microbes and viruses such as HIV and CMV [15,16,26,27,28]. Our data indicate that HPP would be a more suitable method to make human milk safe over thermal pasteurization, as indicated by the retention of functional IgAs in human milk, and the retained neutralization towards the clinical SARS-CoV-2 isolate.

This study has several strengths. First, we were able to collect clinical data, blood and human milk from almost all participants. Second, by using different methods to measure antibodies in serum and human milk we created a robust dataset, capturing the variation in antibody responses. As we included women with varying time frames between the onset of COVID-19 symptoms and collecting samples, we were able to investigate antibody dynamics within each individual subject. Moreover, by using virus neutralizing as a functional readout, we were able to draw conclusions on the effectiveness of the antibodies against the virus.

Human milk is known to be a safe product that can be used for preventive strategies, especially compared to pharmaceutical interventions (either medication or vaccination), and no detrimental side effects are to be expected from its intake. In a recent preprint study, it was determined that the virus itself is not present in human milk and that even if there is detected virus on the breast skin, viral contamination is effectively removed by cleaning the breast before pumping [29]. With breast cleaning and pumping strategies and by pasteurizing human milk it is possible to provide a safe product for donor human milk banks to use for vulnerable populations. Furthermore, neutralizing antibodies against the SARs-CoV-2 virus could be extracted from human milk and used as a highly targeted COVID-19 therapeutic.

However, using human milk as a preventive strategy requires ample availability of human milk from COVID-19 recovered women. While rates of seroprevalence of anti-SARS-CoV-2 IgG antibodies in the general population varies widely, “milk prevalence” rates in pregnant and lactating women are not known and might differ substantially among mothers from region to region.

The possibility to purify IgA from human milk from mothers who have recovered from COVID-19 also opens possibilities, but again requires availability of sufficient seroconverted milk. Currently major issues with obtaining donated milk, includes limitations of donors to visit milk banking facilities, many of which are associated with hospitals, of which the general population has limited access [30]. Even though we are still far away from clinical applications, efforts should be undertaken to investigate the possibility of using human milk antibodies as a preventive strategy against SARS-CoV-2 infection and subsequent spread.

## 5. Conclusions

Human milk of mothers who were previously infected with SARS-CoV-2 contained significant amounts of IgA against SARS-CoV-2 for at least 13 weeks after the onset of symptoms. After pasteurization, total IgA antibody levels were affected by HoP, while SARS-CoV-2 specific antibody levels were affected by HPP. Human milk samples of several donors had neutralization capacity against a pseudovirus of SARS-CoV-2, which remained after non-thermal pasteurization. No correlation was observed between human milk antibody levels and neutralization capacity.

## Figures and Tables

**Figure 1 nutrients-13-01645-f001:**
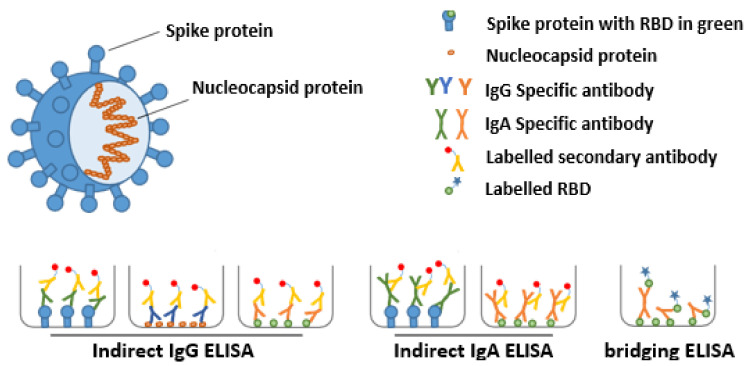
Schematic representation of SARS-CoV-2 and the different ELISA assays used to detect SARS-CoV-2-reactive antibodies. The spike (including the receptor binding domain (RBD)) and nucleocapsid proteins of SARS-CoV-2 are depicted in the context of the virus. SARS-CoV-2 specific antibodies were detected using multiple complementary ELISA assays. The indirect ELISA assays using S, RBD or N were used to detect IgG or IgA specific-antibodies (green, blue or orange, respectively) and the bridging ELISA assay was used to detect total Ig against the RBD.

**Figure 2 nutrients-13-01645-f002:**
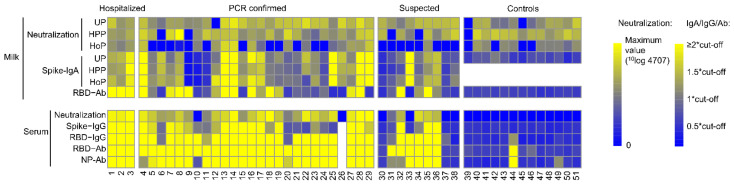
Multiple assay assessment of SARS-CoV-2 antibody levels in human milk and serum. Colors from blue to yellow indicate increasing levels of antibodies in milk and serum, relative to the cut-off values of the respective assays, with all levels over 2 times the cut-off being bright yellow, as indicated by the color scale. For the neutralization of the pseudovirus by milk (unpasteurized (UP), high pressure pasteurized (HPP) and holder pasteurized (HoP)) or serum, colors from blue to yellow indicate increasing neutralization capacity. Ab; total Ig. * multiplication sign.

**Figure 3 nutrients-13-01645-f003:**
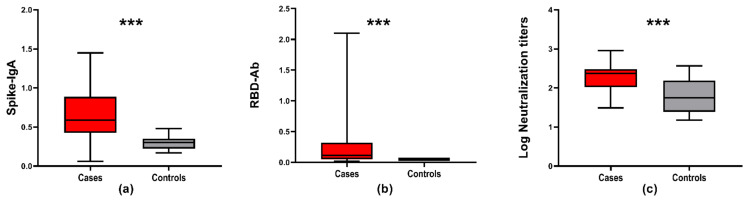
Assessment of human milk antibody interaction with SARS-CoV-2 proteins in unpasteurized milk from the case and control groups. All box plots depict the interquartile range (IQR) as a median value with a lower 25th and upper 75th quartile range, and lower and upper whiskers to indicate the variability outside of the IQR. All cases are depicted in red and all controls are depicted in grey. *** indicates a *p*-value < 0.001 (**a**) OD450 nm SARS-CoV-2 Spike-IgA (*p* < 0.001), (**b**) OD450 nm SARS-CoV-2 RBD-Ab (*p* < 0.001) and (**c**) neutralization titers in unpasteurized milk of the cases and controls (*p* < 0.001).

**Figure 4 nutrients-13-01645-f004:**
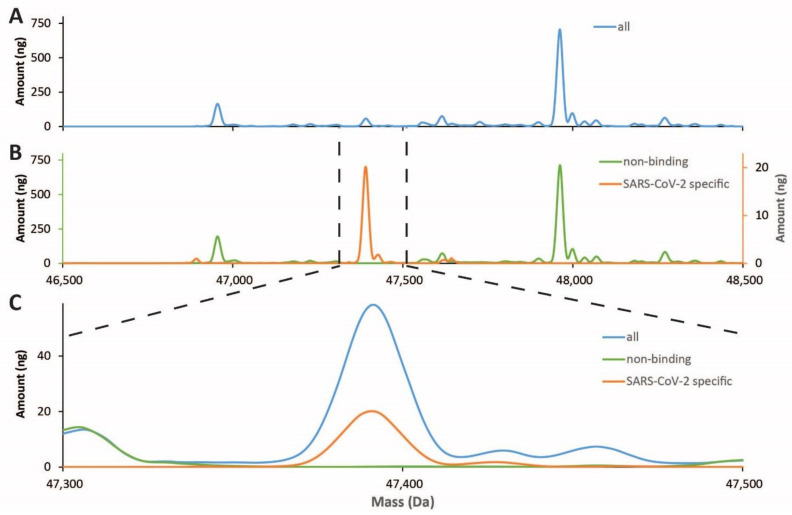
LC-MS profiles of Fab fragments originating from IgA clones present in human milk. (**A**) Profile of IgA clones detected in the unpasteurized milk of Patient 1; (**B**) SARS-CoV-2 antigen-specific affinity purification yields specific clones (orange line; right *y*-axis), distinct from the depleted flow-through fraction (green line, left *y*-axis); (**C**) Overlay of the three profiles reveals the high specificity of a few antigen-specific clones.

**Figure 5 nutrients-13-01645-f005:**
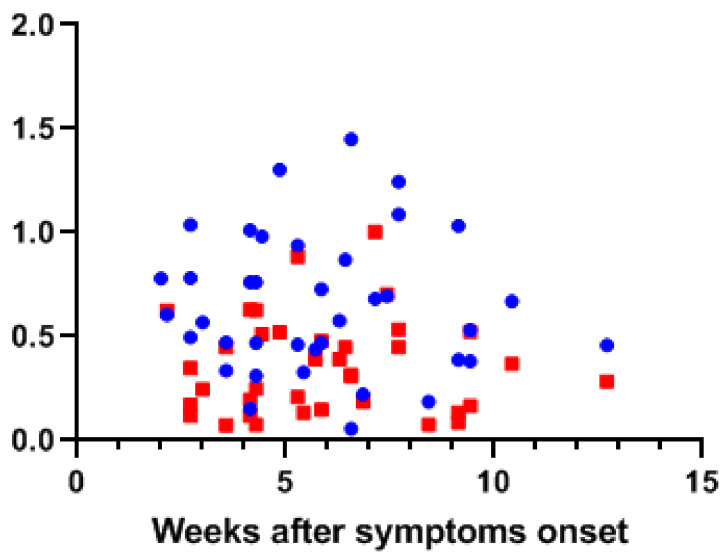
Detection of antibody levels in human milk and serum relative to onset of COVID-19 symptoms. The OD_450 nm_ values for human milk spike protein IgA (blue dots) and serum IgG (red squares) levels from ELISA are plotted against the sampling time point in weeks after the onset of COVID-19 symptom.

**Figure 6 nutrients-13-01645-f006:**
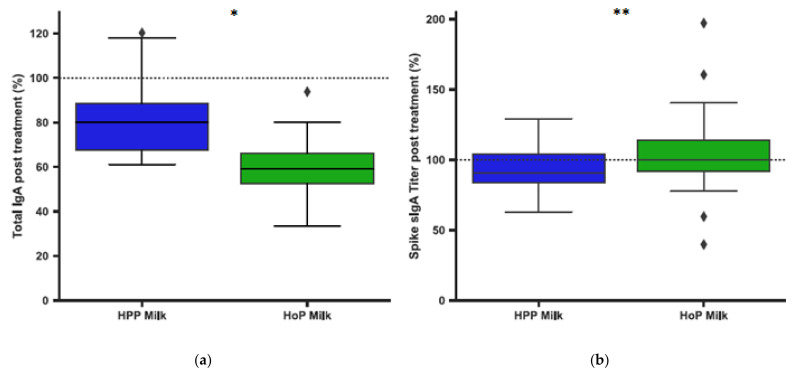
Assessment the effects of pasteurization on human milk IgA levels. All figures are shown as box and whisker plots as median and IQR for human milk expressed as the percentage of treated relative to untreated human milk. ◆ are values outside the IQR. The * indicates a *p*-value < 0.05, the ** a *p*-value < 0.01. (**a**) IgA retention according to LC-MS profiles following HoP (*n* = 9) and HPP (*n* = 9) (*p* = 0.020) (**b**) Spike IgA titers following HoP (*n* = 38) and HPP (*n* = 38) (*p* = 0.006).

**Figure 7 nutrients-13-01645-f007:**
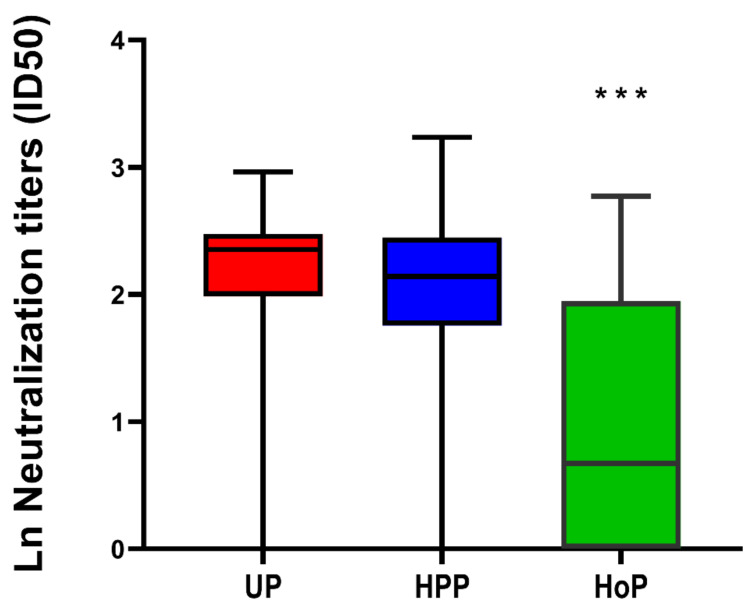
SARS-CoV-2 virus neutralization in unpasteurized and pasteurized human milk. The median and interquartile ranges of neutralizing activity against the pseudovirus, expressed as ln neutralizing capacity (50% inhibitory dilution), for unpasteurized (UP) milk, high pressure pasteurized (HPP) milk and Holder pasteurized (HoP) milk of PCR positive and suspected participants (*n* = 38). Neutralization capacity was generally preserved after the HPP pasteurization (*p* = 0.906) and not after HoP (*p* < 0.001) ***, relative to UP milk.

**Table 1 nutrients-13-01645-t001:** Patient characteristics of the lactating women with a confirmed or highly probable COVID-19 and controls.

Characteristics	Confirmed COVID-19 *N* = 29		Suspected COVID-19 *N* = 9		Controls *N* = 13
Gestational age—weeks median (IQR)	39.7 (38.5, 40.7)		38.8 (36.8, 40.4)		40.7 (39.6, 41.1)
Age of child—weeks median (IQR)	28.9 (12.1, 39.5)		12.6 (8.4, 40.7)		6.1 (4.3, 7.4)
Age of mother—years mean (SD)	31.1 (3.1)		30.3 (4.1)		33.2 (3.3)
Time between start of clinical symptoms and collection of human milk—weeks mean (SD)	5.9 (2.6)		5.7 (2.1)		NA
Symptoms and duration in days	*N*. (%)	Median (IQR)	*N*. (%)	Median (IQR)	
Fever > 37.5 °C	21 (72%)	3 (1, 5)	6 (67%)	1 (0, 4)	NA
Cold	24 (83%)	12 (5, 20)	4 (44%)	7 (4, 54)	NA
Cough	21 (72%)	14 (5, 28)	5 (56%)	6 (3, 12)	NA
Sore throat	21 (72%)	6 (4, 14)	5 (56%)	6 (3, 12)	NA
Tachypnea	5 (17%)	11 (4, 14)	2 (22%)	NA	NA
Dyspnea	14 (38%)	7 (3, 28)	1 (11%)	NA	NA
Stomachache	5 (17%)	2 (1, 9)	2 (22%)	NA	NA
Nausea	5 (17%)	3 (2.5, 14)	1 (11%)	NA	NA
Vomiting	2 (7%)	NA	0	NA	NA
Diarrhea	5 (17%)	2 (1, 25)	3 (33%)	NA	NA
Headache	24 (83%)	5 (2, 12)	8 (89%)	7 (4, 14)	NA
Photophobia	2 (7%)	NA	0	NA	NA
Anosmia	18 (62%)	20 (13)	6 (67%)	9 (4, 22)	NA
Ageusia	17 (59%)	19 (13)	4 (44%)	14 (11, 25)	NA
Fatigue	24 (83%)	20 (16)	8 (89%)	10 (4, 36)	NA
Anorexia	10 (34%)	12 (8, 21)	3 (33%)	NA	NA
Hospital admission	3 (10%)	NA	0	NA	NA
Non-invasive respiratory support (O2) during admission	2 (7%)	NA	0	NA	NA

IQR = interquartile range, SD = standard deviation, NA = not applicable.

## Data Availability

Data is contained within the article or Appendix A. The data presented in this study are available in (Appendix A).

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
