# Peer review of "Human Milk from Previously COVID-19-Infected Mothers: The Effect of Pasteurization on Specific Antibodies and Neutralization Capacity"

_nutrients, 2021, doi:10.3390/nu13051645_

Round 1

Reviewer 1 Report

I thank the authors for the new analysis they performed and representation of the data which have significantly improved the quality of the manuscript (and changed some key messages).

I have additional comments here below:

Title: Human milk; a source of SARS-CoV-2 specific sIgA antibodies,  highly stable after pasteurization

  • I do not see this as the main message of the manuscript.

To me the messages of the paper are:

  • Milk from infected mother contain Abs
  • The milk neutralising activity of a pseudovirus does not correlate with levels of Abs
  • HoP but not HPP affect total IgA Ab levels. HPP but  not HoP affect Spike IgA levels ( please change the abstract)
  • HPP maintains milk neutralising activity of a pseudovirus but not HoP

=> Please choose  the main message you want to convey and adapt the title and conclusion of the abstract accordingly

Abstract:

  • “We aimed to determine the presence and neutralization capacity of SARS-CoV-2 antibodies in human milk of mothers who recovered from COVID-19” => the study is not designed to address this question ( requires Antibody depletion experiments)
  • “ Human milk contained abundant SARS-CoV-2 antibodies in 83% of the proven cases, in 67% of the suspected 35 cases and in none of the controls” => the authors can’t conclude there is no detected Antibody in controls as they use the control to define the cut off. Same comment for result section.

Results

  • Please show the data for all the antibodies tested with graphs similar as shown in supplementary Fig 1 for Spike IgA. Indicate on the graph the cut off that was used. These data need to be shown in a paper which objective is to describe the presence of Abs in milk of infected mothers. Please integrate those graphs in the main manuscript.
  •  “Spike protein sIgA and IgG levels (OD450) in human milk (blue dots) and serum (red squares) using ELISA, and  sampling time point in weeks after symptom onset”  => I guess the authors mean IgA in breast milk and IgG in serum, please correct.
  • “In contrast, we observe a significant difference in the levels of IgA anti S protein antibodies before and after the two types of pasteurization, with lower levels in HPP milk compared to HoP milk (Figure 5b and Supplemental Figure 2)”. => This novel info might need to be added to the abstract
  • Fig. 6, nice figure! => Please add 3 asterix on HoP column
  • Neutralisation activity section

“2.36 (IQR 2.03-2.53) compared to milk from women with a suspected infection 2.37 (IQR 1.96-2.39) or controls 1.75 (IQR 1.39-2.19) 330 (p=0.009). Neutralization capacity differed between unpasteurized and pasteurized milk.” => Please indicate what those numbers are .

  • replication inhibition with a SARS-CoV-2 clinical isolate => please show the neutralisation titres as shown for pseudovirus assay in a supplementary figure
  • “ Remarkably, some (n=3) HPP pasteurized milk samples did show neutralizing capacity while the same sample before pasteurization did not 340 exert such an effect.” => Is this remarkable or does it question the validity of the assay as indicated in the discussion?

Discussion

  • Please start by summarising the main points.
  • “Neutralizing capacity was also observed in milk samples of the control group, which can be explained by the presence of other antiviral proteins, besides antibodies, such as lactoferrin [23]. => I feel worried that the authors who write a paper on neutralising activity oh human milk do not refer correctly to the paper of Fan et al that shows that Lactoferrin is probably NOT responsible for neutralising activity of human milk
  • Conclusion, => please add the conclusion brought by pseudovirus assay and you may say this is supported by preliminary data with clinical isolates

Author Response

Reviewer 1

I thank the authors for the new analysis they performed and representation of the data which have significantly improved the quality of the manuscript (and changed some key messages).

Thank you for your nice comment and taking the time to review our article again. According to your suggestions, we have made additional changes to our manuscript.

I have additional comments here below:

Title: Human milk; a source of SARS-CoV-2 specific sIgA antibodies, highly stable after pasteurization

  • I do not see this as the main message of the manuscript.

To me the messages of the paper are:

  • Milk from infected mother contain Abs
  • The milk neutralising activity of a pseudovirus does not correlate with levels of Abs
  • HoP but not HPP affect total IgA Ab levels. HPP but  not HoP affect Spike IgA levels ( please change the abstract)
  • HPP maintains milk neutralising activity of a pseudovirus but not HoP

=> Please choose the main message you want to convey and adapt the title and conclusion of the abstract accordingly

Reply: thank you for your comment. We changed the title into: “SARS-CoV-2 specific antibodies in human milk from previously infected mothers; the effect of pasteurization and neutralization capacity.” Moreover, we have adjusted our conclusion of the abstract into: “Human milk of COVID-19 recovered mothers contains SARS-CoV-2 specific antibodies and may represent a safe and effective immunization strategy after high pressure pasteurization.”

We also adjusted our results. We removed that “after pasteurization of the milk SARS-CoV-2 antibodies were detected with both methods of pasteurization” and added “After pasteurization, total IgA antibody levels were affected by HoP, while SARS-CoV-2 specific antibody levels were affected by HPP” as suggested by the reviewer.

Abstract: 

  • “We aimed to determine the presence and neutralization capacity of SARS-CoV-2 antibodies in human milk of mothers who recovered from COVID-19” => the study is not designed to address this question (requires Antibody depletion experiments)

 Reply: Thank you for your comment. We adjusted our aim into: “We aimed to determine the presence and pseudovirus neutralization capacity of SARS-CoV-2 specific IgA in human milk of mothers who recovered from COVID-19 and the effect of pasteurization on these antibodies.”

 “Human milk contained abundant SARS-CoV-2 antibodies in 83% of the proven cases, in 67% of the suspected 35 cases and in none of the controls” => the authors can’t conclude there is no detected Antibody in controls as they use the control to define the cut off. Same comment for result section.

Reply: we removed the statement on the antibodies in the control group in both abstract and result section. 

Results

  • Please show the data for all the antibodies tested with graphs similar as shown in supplementary Fig 1 for Spike IgA. Indicate on the graph the cut off that was used. These data need to be shown in a paper which objective is to describe the presence of Abs in milk of infected mothers. Please integrate those graphs in the main manuscript.

Reply: Thank you for this nice suggestion. We added a figure on RBD antibodies in human milk as well and replaced the figure to the main manuscript (currently figure 3).

 “Spike protein sIgA and IgG levels (OD450) in human milk (blue dots) and serum (red squares) using ELISA, and sampling time point in weeks after symptom onset”=> I guess the authors mean IgA in breast milk and IgG in serum, please correct.

Reply: thank you for this observation. We corrected it.

 “In contrast, we observe a significant difference in the levels of IgA anti S protein antibodies before and after the two types of pasteurization, with lower levels in HPP milk compared to HoP milk (Figure 5b and Supplemental Figure 2)”. => This novel info might need to be added to the abstract

Reply: as described in our reply on your comments in the abstract, we have added this information.

  • 6, nice figure! => Please add 3 asterix on HoP column

Reply: thank you for your nice comment, we have added the asterix.

 Neutralisation activity section

“2.36 (IQR 2.03-2.53) compared to milk from women with a suspected infection 2.37 (IQR 1.96-2.39) or controls 1.75 (IQR 1.39-2.19) 330 (p=0.009). Neutralization capacity differed between unpasteurized and pasteurized milk.” => Please indicate what those numbers are.

Reply: we added this information to the result section “In both cases (proven and suspected) and controls, neutralization capacity was observed in unpasteurized milk, although, the median of the ln transformed neutralizing capacity defined as 50% inhibitory dilution, was higher in milk from women with proven COVID-19 2.36 (IQR 2.03-2.53) compared to milk from women with a suspected infection 2.37 (IQR 1.96-2.39) or controls 1.75 (IQR 1.39-2.19) (p=0.009).”

  • replication inhibition with a SARS-CoV-2 clinical isolate => please show the neutralisation titres as shown for pseudovirus assay in a supplementary figure

Reply: we added a figure on the neutralization titers as supplemental figure 2.

  • “Remarkably, some (n=3) HPP pasteurized milk samples did show neutralizing capacity while the same sample before pasteurization did not 340 exert such an effect.” => Is this remarkable or does it question the validity of the assay as indicated in the discussion?

Reply: the fact that some HPP pasteurized milk samples did show neutralizing capacity while the same sample before pasteurization did not exert such an effect does indeed question the validity of the assay as indicated in the discussion. We removed “remarkably” in the result section.

Discussion

  • Please start by summarising the main points.

Reply: thank you for this suggestion. We summarized our main findings in the first paragraph of the discussion and discuss our findings further on.

 First paragraph is now:  We demonstrate that human milk of mothers who recovered from COVID-19 contains significant amounts of IgA against SARS-CoV-2, for at least 13 weeks following the onset of COVID-19 symptoms. After pasteurization, total IgA antibody levels were affected by HoP, while SARS-CoV-2 specific antibody levels were affected by HPP. Pseudovirus neutralizing capacity of the human milk samples was only retained with the HPP approach. No correlation was observed between milk antibody level and neutralization capacity.

  • “Neutralizing capacity was also observed in milk samples of the control group, which can be explained by the presence of other antiviral proteins, besides antibodies, such as lactoferrin [23]. => I feel worried that the authors who write a paper on neutralising activity oh human milk do not refer correctly to the paper of Fan et al that shows that Lactoferrin is probably NOT responsible for neutralising activity of human milk

Reply: thank you for this comment. Indeed, this study showed that Lactoferrin has limited inhibition effects and that other components also should play an important role in neutralizing activity of human milk. We deleted this example from the discussion.

 Conclusion, => please add the conclusion brought by pseudovirus assay and you may say this is supported by preliminary data with clinical isolates

Reply: We adjusted our conclusion into: “Human milk of mothers who were previously infected with SARS-CoV-2 contained significant amounts of IgA against SARS-CoV-2 for at least 13 weeks after the onset of symptoms. After pasteurization, total IgA antibody levels were affected by HoP, while SARS-CoV-2 specific antibody levels were affected by HPP. Milk samples of several donors had neutralization capacity against a pseudovirus of SARS-CoV-2, which remained after non-thermal pasteurization. No correlation was observed between milk antibody level and neutralization capacity.”

Reviewer 2 Report

Nice work on improving the manuscript.

Author Response

Reviewer 2

Nice work on improving the manuscript.

Reply: thank you for your nice complements and taking the time to review our manuscript again.

Round 2

Reviewer 1 Report

I thank the authors for their modifications. I just have 2 details to add as indicated here below. I am very pleased to have contributed to the publication of these important findings! Title SARS-CoV-2 specific antibodies in human milk from previously infected mothers; the effect of pasteurization and neutralization capacity I would suggest to be clearer: “Human milk from SARS-CoV-2 previously infected mothers: the effect of pasteurization on specific antibodies and neutralization capacity “ Graphs: Please use asterisks on all graphs to indicate p value instead of numbers and use the classical definition * p

Author Response

Reviewer 1

I thank the authors for their modifications. I just have 2 details to add as indicated here below. I am very pleased to have contributed to the publication of these important findings! Title SARS-CoV-2 specific antibodies in human milk from previously infected mothers; the effect of pasteurization and neutralization capacity I would suggest to be clearer: “Human milk from SARS-CoV-2 previously infected mothers: the effect of pasteurization on specific antibodies and neutralization capacity “ Graphs: Please use asterisks on all graphs to indicate p value instead of numbers and use the classical definition *

Reply: Thank you for reviewing our manuscript and your constructive suggestions which improved our paper. We have adjusted our title into: “Human milk from SARS-CoV-2 previously infected mothers: the effect of pasteurization on specific antibodies and neutralization capacity”. Moreover, we have used asterisks to indicate p value on all graphs if applicable.

This manuscript is a resubmission of an earlier submission. The following is a list of the peer review reports and author responses from that submission.

Round 1

Reviewer 1 Report

Van Keulen et al., conducted a well-designed study with a series of experiments showing that human milk from women who had recovered from covid-19 contained an abundance of SARS-CoV-2 antibodies that were capable of neutralizing SARS-CoV-2 and a pseudovirus. Furthermore, this group showed that different pasteurization methods resulted in the presence of SARS-CoV-2 antibodies but differences in virus neutralizing capacity. These data show that human milk from infected mothers offers protection to infants against this virus but the protection is influenced by heat processing which is critical for donor milk banks to address in the time of the pandemic.

This reviewer has some minor comments.

  • Breast milk and Breastmilk are inconsistently used throughout. Please change to “Breast milk” or “human milk” as two words throughout the entire manuscript.

  • Line 189. It’s not clear that you applied the two different pasteurization techniques to all samples for a thorough comparison between matching raw milk, vs. milk treated by HoP vs. HPP methods.

  • It would be very interesting to show the correlation coefficient and p values for the between normalized antibody concentrations and weeks after symptom onset shown in figure 4.

  • Were there statistical differences between HoP and HPP shown in Figure 5a? If so, please report p values.

  • Please provide your thoughts in the discussion section explaining why IgA in so many unpasteurized human milk samples were not effective in targeting the S protein of SARS-CoV-2 shown in Figure 6.

Author Response

Reviewer 1:

Van Keulen et al., conducted a well-designed study with a series of experiments showing that human milk from women who had recovered from covid-19 contained an abundance of SARS-CoV-2 antibodies that were capable of neutralizing SARS-CoV-2 and a pseudovirus. Furthermore, this group showed that different pasteurization methods resulted in the presence of SARS-CoV-2 antibodies but differences in virus neutralizing capacity. These data show that human milk from infected mothers offers protection to infants against this virus but the protection is influenced by heat processing which is critical for donor milk banks to address in the time of the pandemic.

This reviewer has some minor comments.

Breast milk and Breastmilk are inconsistently used throughout. Please change to “Breast milk” or “human milk” as two words throughout the entire manuscript. 

Thank you for noticing this. We have replaced breastmilk and breast milk by human milk throughout the entire manuscript to make it consistent.

Line 189. It’s not clear that you applied the two different pasteurization techniques to all samples for a thorough comparison between matching raw milk, vs. milk treated by HoP vs. HPP methods. 

We clarified in the method section that we applied two different pasteurization techniques to evaluate the amount of SARS-CoV-2 antibodies and neutralizing capacity (lines 197-200). 
It would be very interesting to show the correlation coefficient and p values for the between normalized antibody concentrations and weeks after symptom onset shown in figure 4. 

Thank you for this comment. We agree with the reviewer that it would be very interesting to elucidate the relation between antibody levels and the time after onset of symptoms. However, this could be influenced by several other factors such as for example disease severity. Unfortunately, we are not able to adjust for these covariates in this small size study. At this moment, we are collecting milk of thousands of women to further evaluate this and we will certainly take this interesting suggestion into account in that manuscript.
Were there statistical differences between HoP and HPP shown in Figure 5a? If so, please report p values.

There were statistical differences. We added the p value (p=0.027) to the figure.

Please provide your thoughts in the discussion section explaining why IgA in so many unpasteurized human milk samples were not effective in targeting the S protein of SARS-CoV-2 shown in Figure 6.

We agree with the reviewer that this is a remarkable result. We have added our thoughts on this in the discussion (lines 338-340).It is good to observe that similar results were found in the publication from last week: mBio . 2021 Feb 9;12(1):e03192-20. doi: 10.1128/mBio.03192-20. In this report, as in our study approximately 80% of the pCR positive mothers excreted spike IgA against SARS-CoV-2 in human milk.

Reviewer 2 Report

The paper of van Keulen et al addresses a very important question in the context of child nutrition during COVID 19 pandemics: the protective effects of breast milk including the influence of milk pasteurisation by 2 different methods.

The authors have a fantastic set of data but at this stage, the manuscript does not allow to get a proper analysis, understanding and interpretation of the data because of a lack of clarity in the methods, the description of the control  for the tests and the representation of the data. 

 Detailed comments

Abstract

Please modify according to comments in the different sections.

Methods

Evaluation of antibodies in the serum and breast milk

  • The description of the Elisa requires more clarity and be more precise:

Line 120 “ELISA with the SARS-CoV-2 spike protein for IgA and IgG”. This sentence is very hard to understand.  Do you mean: Plates were coated with SARS-CoV-2 spike to detect SARS-CoV-2 spike specific IgA  in milk and SARS-CoV-2 spike -specific IgG in serum. If yes, please make change for the whole paragraph. 

 Line 123 “bridging ELISA with the SARS-CoV-2 RBD and N protein for total Ig”: what do you mean by total Ig?  What is this Elisa bringing as compared to the indirect one?

Line 135 “ blocking step with casein and dilution in casein”: which % of casein and in what buffer?

Line 141. Is this a sub-heading?

Line 142 “ total antibody Elisa”, please be more precise

Line 145: Biotinilinated  RBD, provider? Home made?

Line 147: monoclonal IgG: provider?

  • Importantly, which negative control were used ? how was the cut-off for detection determined ?

Neutralisation assay

Linle 165: what dose of pseudovirus was added?

Line 173: “breast milk with SARS-Cov2 possess” a word is missing

 Results

I find the Figure 2 very hard to read and it does not allow a fine analysis and interpretation of the data.  Could the authors provide Figures with individual plots for each mother for each Elisa assay as well as for the neutralisation assay?

As the author have the data, a figure with an analysis of the correlation between IgG in serum and IgA in milk would be informative.

Figure 6 is not ideal to analyse the data neither. I would split this figure in 2. One figure where the authors show the association between IgA level ( and their specificity) and neutralisation activity and another figure with the effect of heating on neutralisation activity, including control samples .

It is not clear why the authors use two different neutralisation assays in figure 2 and figure 6, please explain. There is also a need to discuss why results are different with 2 assays.

Discussion

Line 323. Please cite the work of Fan et al,  “The effect of whey protein on viral infection and replication of SARS-CoV-2 and pangolin coronavirus in vitro, that describes the effect of whey protein in inhibiting viral infection and shows Lactoferrin has minimal activity”.

Author Response

Reviewer 2:

The paper of van Keulen et al addresses a very important question in the context of child nutrition during COVID 19 pandemics: the protective effects of breast milk including the influence of milk pasteurisation by 2 different methods.

The authors have a fantastic set of data but at this stage, the manuscript does not allow to get a proper analysis, understanding and interpretation of the data because of a lack of clarity in the methods, the description of the control  for the tests and the representation of the data. 

With help of the constructive remarks of this reviewer, we think we improved the manuscript significantly.

Detailed comments

Abstract

Please modify according to comments in the different sections.

We have adjusted the abstract based on the comments below.
Methods

Evaluation of antibodies in the serum and breast milk

The description of the Elisa requires more clarity and be more precise:

Line 120 “ELISA with the SARS-CoV-2 spike protein for IgA and IgG”. This sentence is very hard to understand.  Do you mean: Plates were coated with SARS-CoV-2 spike to detect SARS-CoV-2 spike specific IgA  in milk and SARS-CoV-2 spike -specific IgG in serum. If yes, please make change for the whole paragraph. 

We would like to thank the reviewer for this comment. We have clarified this through the whole paragraph.

Line 123 “bridging ELISA with the SARS-CoV-2 RBD and N protein for total Ig”: what do you mean by total Ig?  What is this Elisa bringing as compared to the indirect one?

We have clarified this, by adding that we mean specific total Ig (IgG, IgM and IgA in human milk and serum) (lines 120-126)

Line 135 “ blocking step with casein and dilution in casein”: which % of casein and in what buffer?

The % of casein is 1% and the buffer is PBS. We have added this in line 138.

Line 141. Is this a sub-heading?

We have adjusted it into a sub-heading according the journal style (line 143).

Line 142 “ total antibody Elisa”, please be more precise

We have clarified this. We have adjusted it into total Ig (IgG, IgA and IgM) (lines 145-146).

Line 145: Biotinilinated  RBD, provider? Home made?

We have added information regarding our materials in the method section. Our materials are in-house made, as explained in the reference.

Line 147: monoclonal IgG: provider?

We have added information regarding our materials in the method section. Our materials are in-house made, as explained in the reference.

Importantly, which negative control were used ? how was the cut-off for detection determined ?

In order to determine the cut-off value, we used pre-pandemic controls (lines 154-155).

Linle 165: what dose of pseudovirus was added?

For our pseudovirus analysis, approximately 1 × 103 infectious units were added. We have added this in the manuscript (line 171).

Line 173: “breast milk with SARS-Cov2 possess” a word is missing

Thank you for noticing. We have added the word “antibodies” to this sentence (line 180) and change breast milk  into human milk

 Results

I find the Figure 2 very hard to read and it does not allow a fine analysis and interpretation of the data. Could the authors provide Figures with individual plots for each mother for each Elisa assay as well as for the neutralisation assay?

We regret that the figure is hard to read for the reviewer. Since we have conducted several experiments within each of the 38 participants it would be too extensive to show the results in individual plots. However, we agree with the reviewer that it is interesting to show the results for each participant and therefore, in our opinion, a heatmap was the best option to show our results. 
As the author have the data, a figure with an analysis of the correlation between IgG in serum and IgA in milk would be informative.

Thank you for this comment. It would indeed be very interesting to elucidate the correlation between IgG in serum and sIgA in milk. However, these results would be hard to interpret with this sample size. However, as also mentioned in the rebuttal of the previous reviewer, we will address this issue in the next manuscript in which we will be describing >1500 serum-milk pairs. 

Figure 6 is not ideal to analyse the data neither. I would split this figure in 2.
One figure where the authors show the association between IgA level ( and their specificity) and neutralisation activity and another figure with the effect of heating on neutralisation activity, including control samples .

We have considered different approaches to show these results. Our most important finding is that virus neutralization capacity of SARS-CoV-2 antibodies in breast milk was only retained with the HPP approach and not with HoP. This could be distracted from this figure, since no neutralization capacity was seen for the samples treated with HoP (green dots).

It is not clear why the authors use two different neutralisation assays in figure 2 and figure 6, please explain. There is also a need to discuss why results are different with 2 assays.

Thank you for this valuable comment. In order to assess the neutralizing capacity of breast milk antibodies we first tested this with a clinical isolate. During this procedure, breast milk samples were diluted in order to prevent cell toxicity. This dilution could have influenced the neutralizing capacity of our samples. Therefore we decided to also test the neutralization capacity by an additional approach. We explained that in the discussion section as well in lines 338-340.

Discussion

Line 323. Please cite the work of Fan et al,  “The effect of whey protein on viral infection and replication of SARS-CoV-2 and pangolin coronavirus in vitro, that describes the effect of whey protein in inhibiting viral infection and shows Lactoferrin has minimal activity”.

We have added this reference to our discussion.

Round 2

Reviewer 2 Report

I thank the authors for answering the comments. There are still major points that have not been solved.

Methods

- Major comments

Negative control and cut off. The authors refer to a previous publication (18) for the Elisa used in the submitted manuscript. However, here, the authors use human milk and not serum and I expect there might be a lot of non-specific binding as observed in previous publication ( Fox et al, https://doi.org/10.1016/j.isci. ). It is therefore important to state here too what negative control was used and how cut off for positive samples was defined for all the Elisa (not only the bridge Elisa).

- Minor comment

Line 158. “Elisa with SARS cov2 protein” please modify “Detection of anti-SARS CoV2 Ig in serum and human milk”

Results

- Major comment

Figure 2. I reiterate here that it is necessary the authors show individual data ( or boxes with errors bars) on separate plots for each assay. This is required to have quantitative analysis of the data and perform statistical analysis on the changes the authors observed with pasteurisation. No conclusion on the significance of the data can be drawn without statistical analysis.

Figure 6. same comment as in previous review and same comment as in Figure 2. There is a need for statistical analysis in order to be able to conclude on the data.

- Minor comment

Line 298: “None of the 13 controls expressed SARS-CoV-2-reactive antibodies in human milk” . This sentence does not make sense as the authors have used the pre-pandemic control to establish the cut off. Thus I would delete this sentence or modify.

Author Response

I thank the authors for answering the comments. There are still major points that have not been solved.

Thank you for your final comments. We hope that the changes made in this revised paper are sufficient. Please find below the adjustments we have made in response to your comments. 

Methods

- Major comments

Negative control and cut off. The authors refer to a previous publication (18) for the Elisa used in the submitted manuscript. However, here, the authors use human milk and not serum and I expect there might be a lot of non-specific binding as observed in previous publication ( Fox et al, https://doi.org/10.1016/j.isci. ). It is therefore important to state here too what negative control was used and how cut off for positive samples was defined for all the Elisa (not only the bridge Elisa).

We would like to thank you for your comment. We used different cut-offs for the different ELISAs. For the ELISA Spike-protein the healthy controls are used in order to determine the cut-off value. For the RBD and N-protein assay, the data of pre-pandemic controls were used in order to determine the cut-off value. In order to clarify this we have added additional information to the methods in lines 144-147: ‘ The healthy controls (serum and human milk) were used to determine cut-off values defined as the mean plus two times the standard deviation. Specificity of the ELISA was shown to be >95% for both serum and human milk and the sensitivity was >90% for serum and >80% for human milk’.

- Minor comment

Line 158. “Elisa with SARS cov2 protein” please modify “Detection of anti-SARS CoV2 Ig in serum and human milk”

Thank you for your comment. We are not sure which line is meant by the reviewer. We think this comments points to the heading of the paragraph in line 133. We adjusted the heading to ‘Detection of anti-SARS-CoV-2 Ig in serum and human milk with ELISA’ in line 133.

Results

- Major comment

Figure 2. I reiterate here that it is necessary the authors show individual data ( or boxes with errors bars) on separate plots for each assay. This is required to have quantitative analysis of the data and perform statistical analysis on the changes the authors observed with pasteurisation. No conclusion on the significance of the data can be drawn without statistical analysis.

Thank you for this comment. We have added 3 Supplemental Figures with sIgA levels and neutralization capacity for each individual before and after both pasteurization methods in order to quantify the effect of pasteurization. The data can also be found in Supplemental Table 1.
In addition, we performed statistical analyses to test for differences after pasteurization, these results are shown in the paper in:
- Lines: 289-292: However, we did not observe a significant difference in the levels of sIgA anti S protein antibodies before and after the two types of pasteurization (Figure 5b and Supplemental Figure 1).
- Lines 295-298: Figure 5. (a) IgA retention according to LC-MS profiles following holder pasteurization (HoP) and high pressure pasteurization (HPP), expressed as the % relative to untreated human milk (p = 0.027). (b) Spike IgA titers in unpasteurized (UP) milk, high pressure pasteurized (HPP) milk and holder pasteurized (HoP) milk, without significant differences between groups.
- Lines 306-307: In contrast, after HoP a substantial decrease in neutralization capacity was observed (p<0.001).
- Lines 310-312: In 7 of the 38 (18%) unpasteurized milk samples, and in 8 of the 38 (21%) HPP pasteurized milk samples, neutralization capacity was observed (p<0.001). None of the HoP pasteurized milk samples showed neutralization capacity (Supplemental Figure 3).

Moreover, we decided to still include the heatmap in the paper. All individual data is also shown in this figure, in which the color for each individual indicates the values for the different analyses. We agree with the reviewer that separate plots of each assay with individual data adds extra valuable information in order to compare the effects of pasteurization. Therefore, we have added the 3 supplemental figures showing these data.  

Figure 6. same comment as in previous review and same comment as in Figure 2. There is a need for statistical analysis in order to be able to conclude on the data.

As described above, we have added supplemental figures to show the individual data. The individual data regarding antibody level and neutralization capacity is already shown in figure 6 (each dot represents a participant and the colors represents the different pasteurization techniques). In order to compare the effect of different pasteurization methods on neutralization capacity in vitro we have added supplemental figure 2. In addition, in lines 311-312 we stated that a linear correlation was not detected. 

- Minor comment

Line 298: “None of the 13 controls expressed SARS-CoV-2-reactive antibodies in human milk” . This sentence does not make sense as the authors have used the pre-pandemic control to establish the cut off. Thus I would delete this sentence or modify.

We think we caused confusion regarding our included control group, since different control groups were used. We have clarified this, as explained in the major comment of the methods. The control group in this paper is included at the same time as the group with a proven COVID-19 infection, so during the pandemic. The cut-off value for the ELISA assay with Spike protein was determined from our 13 controls, and the cut-off value for the ELISA assay with N protein and RBD was determined from pre-pandemic controls. In order to elucidate this we have made some adjustments throughout the paper, namely the following:
- In lines 107-108 we have added: ‘
at the same time as the group with a proven COVID-19 infection’.
- In lines 157-160 we have added: ‘In order to determine the cut-off values, pre-pandemic controls were used to provide
99% specificity as previously described [18]. The results of the pre-pandemic controls are not described in this paper’.
- In line 242 we have added: during pandemic.

Hopefully these adjustments are sufficient in order to understand the study population.

Round 3

Reviewer 2 Report

I thank the authors for their answers and the changes they have brought. There is however still a lack of scientific rigour in the way the data are presented and interpreted to allow the publication of the manuscript.

This is detailed here below:

Abstract:

“ These antibodies were found capable of neutralizing a clinical  isolate of SARS-CoV-2 and a pseudovirus” this major statement is not supported by the data. The author show the presence of Antibody and neutralisation activity of samples but the causal role is not addressed in this manuscript. Furthermore, they indicate “ We could, however, not detect a linear correlation between milk antibody levels and the virus neutralization capacity (Figure 6)” which is in contradiction with the statement of a role of antibody in neutralising activity

“ virus neutralizing capacity of those antibodies was only retained with the HPP approach”:  as above, it is not demonstrated that neutralising activity is antibody dependent. Data and statistical analysis as they are shown at this stage do not allow to give this conclusion.   

Fig. 4.

Could the author explain what they mean by “normalised” antibody levels?

Fig. 5.

Could the authors add more precise information for this graph:

  • Please indicate what represent the graph: median and IQR?
  • n of samples per group
  • statistical analysis: which test was used? Multiple comparison test? Paired? Parametric? Non parametric? Which software? (this latter info should be added in methods)

Supplemental Fig 2 Neutralization capacity in vitro.

Please give a complete informative title such as “SARS CoV2 virus - neutralising activity of human milk samples “

My understanding is that this is the Supplemental Fig 3 as referred in the text and not the supplemental Fig 2.  

How do the authors explain that a few samples show neutralising activity after HPP but not before? Such as for samples 6 , 7 and 35. This makes me question the validity of the assay.

“ In 7 of 316 the 38 (18%) unpasteurized milk samples, and in 8 of the 38 (21%) HPP pasteurized milk 317 samples, neutralization capacity was observed (p<0.001)”  Which statistical analysis have been performed here?  Which groups are compared?

In their recent publication, Pace et al ( Characterization of SARS-CoV-2 RNA, Antibodies, and Neutralizing Capacity in Milk Produced by Women with COVID-19) found 60% of milk samples have neutralising activity while here only 18% show neutralising activity. How do the authors explain such a difference? This makes me also question the validity of the assay. As indicated in the discussion, the authors had to use very diluted milk and it is quite possible the sensitivity of the assay was then very low.  I would therefore not draw major conclusion on this assay and show it as additional data only.

 Supplemental Figure 3. Neutralization capacity pseudovirus

Please give a complete informative title such as “Pseudo virus  - neutralising activity of human milk samples “.

This is the Supp Fig 2 as referred in the text, please amend

The data of this figure are fundamental for the conclusion of the paper and need to be in the main document. It is necessary to have precise representation of the data including adapted statistical analysis in order to conclude whether HoP/HPP significantly affect neutralising activity.

Please show  

  • a figure with data with median and IQR and n of samples for ctrl and infected mother milk plus stat ( including test that is used)- this one could be supplementary as it is not the main message of the paper
  • a figure with data with median and IQR and n of samples for  UP milk, HPP milk and HoP milk  of participants  plus stat ( including test that is used). This figure has to be in the main doc.

Figure 6. The author conclude “ We could, however, not detect a linear  correlation between milk antibody levels and the virus neutralization capacity (Figure 6).”

Accordingly, could the authors change the conclusion in the abstract as stated above and in the discussion “In cases where high antibody levels were present, both unpasteurized and HPP-treated milk were able to neutralize the virus, whereas HoP-treated milk was not”. This statement is not supported by the data.

Author Response

Comments reviewer

Thank the authors for their answers and the changes they have brought. There is however still a lack of scientific rigour in the way the data are presented and interpreted to allow the publication of the manuscript.

We would like to thank the reviewer for the detailed comments. We hope that the changes made in this revised paper are sufficient.

This is detailed here below:

Abstract:

“ These antibodies were found capable of neutralizing a clinical  isolate of SARS-CoV-2 and a pseudovirus” this major statement is not supported by the data. The author show the presence of Antibody and neutralisation activity of samples but the causal role is not addressed in this manuscript. Furthermore, they indicate “ We could, however, not detect a linear correlation between milk antibody levels and the virus neutralization capacity (Figure 6)” which is in contradiction with the statement of a role of antibody in neutralising activity

Thank you for this comment. We agree with the reviewer that our data is not addressing a causal relationship. Therefore, we have attenuate our results in the abstract section; “Human milk was found to be capable of neutralizing a clinical isolate of SARS-CoV-2 and a pseudovirus in a subset of the samples” in lines 35-37.

“ virus neutralizing capacity of those antibodies was only retained with the HPP approach”:  as above, it is not demonstrated that neutralising activity is antibody dependent. Data and statistical analysis as they are shown at this stage do not allow to give this conclusion.   

We agree that our data does not allow to give this conclusion, as we do not know if the neutralization capacity is effectuated by antibodies or by other human milk compounds. Therefore, we have deleted “those antibodies” and replaced it with “the human milk samples” in line 39.

Fig. 4.

Could the author explain what they mean by “normalised” antibody levels?

We agree with the reviewer that ‘normalized’ raises some questions. In order to clarify this we have made a new figure with the actual spike sIgA and IgG levels in human milk and serum

Fig. 5.

Could the authors add more precise information for this graph:

Please indicate what represent the graph: median and IQR?

Figure 5a. Boxplot showing the median and IQRs (25th, 50th and 75th percentile), the whiskers are Q1 – 1.5 * IQR and Q3 + 1.5 * IQR (this is a standard method for displaying boxplots with outliers). The markers indicate the underlying datapoints, and the outliers (values outside the whiskers) have larger markers.” De software used is: “Python 3.8.8, Pandas 1.2.3(1), Numpy 1.19.2(2), Scipy 1.6.1(3), visualized with Matplotlib 3.3.4(4) and Seaborn 0.11.1”

Figure 5b. Boxplot showing the median and total ranges. De software used is ”Graphpad Prism 8.2.1.”

 n of samples per group

Figure 5a. each group includes 9 samples

Figure 5b. This figure includes all cases. Therefore, the n is 38 for each group.

 statistical analysis: which test was used? Multiple comparison test? Paired? Parametric? Non parametric? Which software? (this latter info should be added in methods)

Figure 5a: for this boxplot a paired sample t test is used. De software used is: “Python 3.8.8, Pandas 1.2.3(1), Numpy 1.19.2(2), Scipy 1.6.1(3), visualized with Matplotlib 3.3.4(4) and Seaborn 0.11.1”

Figure 5b: for this boxplot two paired Wilcoxon signed rank tests are used, one to compare the HPP milk with the unpasteurized milk and one to compare the HoP milk with the unpasteurized milk.  De software used is ”Graphpad Prism 8.2.1.”

We have added the information above in the methods and the results sections.  

Supplemental Fig 2 Neutralization capacity in vitro.

Please give a complete informative title such as “SARS CoV2 virus - neutralising activity of human milk samples “

We adjusted the title into “SARS-CoV-2 neutralizing capacity of human milk of a clinical isolate”

My understanding is that this is the Supplemental Fig 3 as referred in the text and not the supplemental Fig 2.  

The authors accidentally switched the supplemental figures. Thank you for your awareness.

How do the authors explain that a few samples show neutralising activity after HPP but not before? Such as for samples 6 , 7 and 35. This makes me question the validity of the assay.

Thank you for this comment. Our explanation is that the processing might add neutralizing capacity to the milk samples. Which factors play a role is not certain, but as some milk samples that did not contain antibodies against SARS-CoV-2 showed neutralizing capacity, other components in the milk are important as well. We have added this information to the discussion (lines 374-377) as well.

“ In 7 of 316 the 38 (18%) unpasteurized milk samples, and in 8 of the 38 (21%) HPP pasteurized milk 317 samples, neutralization capacity was observed (p<0.001)”  Which statistical analysis have been performed here?  Which groups are compared?

The unpasteurized milk samples were compared to the HPP pasteurized milk samples with a Chi-sqaure test, since the outcome was binary (neutralization yes or no). Since the HoP pasteurized milk samples showed no neutralization capacity we were not able to compare those samples. However, since no neutralization capacity was observed, this group differed per definition. We have added this information to the method section.

In their recent publication, Pace et al ( Characterization of SARS-CoV-2 RNA, Antibodies, and Neutralizing Capacity in Milk Produced by Women with COVID-19) found 60% of milk samples have neutralising activity while here only 18% show neutralising activity. How do the authors explain such a difference? This makes me also question the validity of the assay. As indicated in the discussion, the authors had to use very diluted milk and it is quite possible the sensitivity of the assay was then very low.  I would therefore not draw major conclusion on this assay and show it as additional data only 

Our study demonstrates the neutralization capacity as measured by a clinical isolate assay and with the pseudovirus. The results of these assays show great differences, which could be explained by the differences between these two methods. The clinical isolate assay tests replication inhibition of SARS-CoV-2, and the pseudovirus assay tests the prevention of SARS-CoV-2 infection. We added this information in addition to the information regarding the needed dilution for the clinical isolate assay, in our manuscript in the discussion section in lines (371-374).

Supplemental Figure 3. Neutralization capacity pseudovirus

Please give a complete informative title such as “Pseudo virus  - neutralising activity of human milk samples “.

We have adjusted the title of the figure.

This is the Supp Fig 2 as referred in the text, please amend

We have replaced the figure to the main document, so this comment is not applicable anymore.

The data of this figure are fundamental for the conclusion of the paper and need to be in the main document. It is necessary to have precise representation of the data including adapted statistical analysis in order to conclude whether HoP/HPP significantly affect neutralising activity.

We have replaced this figure to the main document (now Figure 6).

Please show

a figure with data with median and IQR and n of samples for ctrl and infected mother milk plus stat ( including test that is used)- this one could be supplementary as it is not the main message of the paper

Thank you for your suggestion. We have added this figure to the manuscript as supplemental Figure 1 and information on the statistics in the statistical analysis section. Moreover, we have added a sentence on the comparison between infected mothers and controls in lines 271-273.

a figure with data with median and IQR and n of samples for  UP milk, HPP milk and HoP milk  of participants  plus stat ( including test that is used). This figure has to be in the main doc.

 The figure suggested by the reviewer is already in the main document (Figure 5b). The only difference is that we show a total range instead of a IQR. The number of samples and the statistics is now included in the paper (in the caption of the figure and in the method section).

Figure 6. The author conclude “ We could, however, not detect a linear correlation between milk antibody levels and the virus neutralization capacity (Figure 6).”

Accordingly, could the authors change the conclusion in the abstract as stated above and in the discussion “In cases where high antibody levels were present, both unpasteurized and HPP-treated milk were able to neutralize the virus, whereas HoP-treated milk was not”. This statement is not supported by the data.

We agree with the reviewer that our statement is not supported by the data, since a linear correlation between milk antibody levels and the virus neutralization capacity could not be detected. However, the samples with the highest antibody titers were capable of neutralizing the virus, so, it could be suggested that this relationship could not be found due to the small sample size. We have attenuated our statement in the discussion, which is now as the following: “According to the neutralization results, it could be suggested that in cases where high antibody levels were present, both unpasteurized and HPP-treated milk were able to neutralize the virus, whereas HoP-treated milk was not, although, a linear correlation could not be detected.” (lines 363-365). The abstract is already adjusted according to your previous comment.
